involuntary treatment; scoping review; restraint; seclusion; Africa

**Corresponding author:**
Brandon A. Knettel;
Email: Brandon.Knettel@duke.edu

# The management of psychiatric emergencies in Africa: A scoping review of restraint and seclusion practices in clinical settings and their impacts

Shankar Chakkera[1], Julia Sieg[2], Theodora Khofi[3], Rosemina Ayieko[4] and Brandon A. Knettel[4,5,6]

[1]Johns Hopkins University, Baltimore, MD, USA; [2]Faculty of Behavioural and Cultural Studies, Heidelberg University, Heidelberg, Germany; [3]Malawi-Liverpool-Wellcome Trust Clinical Research Programme, Blantyre, Malawi; [4]Duke Global Health Institute, Duke University, Durham, NC, USA; [5]Duke University School of Nursing, Durham, NC, USA and [6]Duke Center for Global Mental Health, Durham, NC, USA

## Abstract

More than 116 million people in Africa live with mental health conditions. However, many African countries lack the infrastructure, training and workforce to effectively manage psychiatric emergencies. This has led to overuse of controversial practices such as physical and chemical restraint and involuntary seclusion, often violating patient rights. We conducted a scoping review of restraint and seclusion practices and their impacts in African clinical settings using the PubMed, Embase, CINAHL, PsycInfo and ProQuest databases. Titles/abstracts and full texts were reviewed for inclusion using the Covidence platform, and 29 studies were included in the final extraction. Restraint and/or seclusion were employed to manage aggression, enable involuntary treatment or prevent self-harm. Patients found restraint and seclusion to be dehumanizing, a cause of posttraumatic stress and a barrier to future help-seeking. Healthcare workers described inadequate training, overuse of restraint and seclusion, injuries and emotional distress after employing these treatments. Further research, intervention development and policy reform are urgently needed to promote humane and patient-centered psychiatric care, including verbal de-escalation training, in underresourced healthcare systems.

## Impact statement

Psychiatric emergencies occur when individuals experience serious mental suffering and behavioral alteration that require immediate treatment. When healthcare workers do not have adequate training or resources to respond effectively, this can lead to violence, discrimination and deprivation of fundamental rights of people facing mental health conditions, including the excessive or unsafe use of interventions such as involuntary restraint and seclusion. Restraint refers to the use of physical, chemical or mechanical methods to restrict a patient's movement, and seclusion is a form of restraint that involves isolating a patient in a room or other space. These topics are critically important in Africa due to the unique challenges facing many health systems on the continent, including resource constraints, stigma surrounding mental health and gaps in the availability of trained mental health professionals. Restraint and seclusion practices, while sometimes necessary to ensure staff and patient safety, raise ethical and human rights concerns that demand urgent attention. We conducted a scoping review of restraint and seclusion practices and their impacts in African clinical settings using the PubMed, Embase, CINAHL, PsycInfo and ProQuest databases, and identified 29 relevant studies on the topic. Restraint and/or seclusion were employed to manage aggression, enable involuntary treatment or prevent harm to oneself or others. Patients found restraint and seclusion to be dehumanizing, a cause of posttraumatic stress and a barrier to future help-seeking. Healthcare workers described inadequate training, overuse of restraint and seclusion, injuries and emotional distress after employing these treatments. This scoping review has illuminated challenges associated with restraint and seclusion across various African contexts, including their overuse, unsafe use and use for punitive purposes. We underscore the need for culturally sensitive and patient-centered care while highlighting the importance of reforming current practices to prioritize safety, dignity and the minimization of trauma. The findings serve as a foundation for guiding policy changes, advocacy efforts and improvements in training for mental health professionals.

## Background

In Africa, more than 116 million people were estimated to be living with mental health conditions before the COVID-19 pandemic, and this burden has likely increased during and after the

pandemic (World Health Organization [WHO], 2022). This situation has been worsened by several factors, including a large gap between the scope of mental health challenges and the number of providers available to provide treatment, the stigma surrounding mental health in many settings and the impact of poverty and conflict situations on lower-income regions of Africa (Singla et al., 2017; Moitra et al., 2022). Although the availability of treatment varies widely across different African settings, there are fewer than two mental health workers per 100,000 people on the continent, with many nations having fewer than one mental health worker per one million people (WHO, 2022). In addition, fewer than 11% of member states in Africa provide pharmacological or psychological interventions at community and primary care levels (WHO, 2022).

For people with serious mental illness (SMI), which is marked by major functional impairment, inadequate mental health treatment leads to increased risk of psychiatric emergencies. Psychiatric emergencies are defined as "serious mental suffering and behavioral alteration, which promptly requires adequate treatment" (Goretti et al., 2023). Often, this involves agitated or aggressive behavior, active psychotic symptoms or other forms of disorganized thinking and behavior. The nature of these symptoms can lead to fear, concern for safety and activation of emergency response systems such as security personnel, law enforcement or medical response. When responders are inadequately resourced and undertrained, their responses are likely to escalate symptoms and worsen the emergency (Baldaçara et al., 2021). This can lead to violence, discrimination and deprivation of fundamental rights of people facing mental health conditions, including the excessive or unsafe use of interventions such as involuntary restraint and seclusion (Alemu et al., 2023).

Restraint refers to the use of physical, chemical or mechanical methods to restrict a patient's movement, and seclusion is a form of restraint that involves isolating a patient in a room or other space. In theory, both methods are intended to be used only when necessary to prevent a patient from harming themself or others, reduce the risk of harm and assist the patient to self-regulate and regain control of their behavior (Parkes and Tadi, 2025). However, throughout history as well as in recent years, there has been widespread documentation of the use of restraint and seclusion in an unsafe, excessive or punitive manner due to lack of resources, education or access to more humane alternatives (Recupero et al., 2011).

Prominent global health agencies have provided limited guidance on the topic of psychiatric restraint and seclusion in recent years, with the most cited guidance coming from the 2006 United Nations Convention on the Rights of Persons with Disabilities (United Nations, 2006) and the WHO's QualityRights Initiative (WHO, 2019). As a result, efforts at advocacy, improvement and reduction of restraint and seclusion practices in Africa have been quite limited and most often occur at a local level (Mutiso and Ndetei, 2024). Involuntary restraint and seclusion are prevalent in many African clinical settings, potentially due to limitations in the human resources and infrastructure needed to prioritize alternative approaches. The use of restraint and seclusion can lead to physical injury, emotional trauma and even death (Mohr et al., 2003). Although the complete abandonment of these practices is difficult to envision, it is critical to minimize their use and to maximize their safety (Khadivi et al., 2004).

In high-income global settings, efforts have included mandated training to improve de-escalation strategies and reduce the use of restraint and seclusion, increased staffing and improved infrastructure (Pérez-Toribio et al., 2022). In addition, legal provisions have been implemented in many countries to help protect patients' safety and rights. For example, the code of practice for the UK Mental

health Act of 1983 states that physical restraint should be used as a last resort where there appears to be a real possibility of harm if withheld. The American Medical Association Code of Medical Ethics also states that restraint should never be for punitive reasons, for convenience or to offset staff shortages (O'Donovan et al., 2023). As a result of these efforts, there have been improvements in the standard of care for psychiatric patients in Western countries over the recent decades, and it will be crucial to implement similar strategies in other global settings, including throughout the continent of Africa.

A 2021 systematic review sought to identify psychiatric hospital reforms in low- and middle-income countries and identified 16 studies on the topic (Raja et al., 2021); only two of the studies were conducted in Africa, and most of the studies described reforms in a single site or region rather than a broader national effort (Uys et al., 1996; Krüger and Lewis, 2011). One more recent effort involved the implementation of WHO QualityRights training in Ghana, which showed positive preliminary engagement. However, no data at the institution-level or patient-level impacts of this training have been reported to date, and the implementers described challenges related to a lack of resources to put the training into action and non-acceptance of guidance to reduce coercive treatments among some providers (Osei et al., 2024). To encourage the success of future efforts, it is vital to gain a clearer understanding of current practices and attitudes related to restraint and seclusion that may hinder future efforts at reform.

The objective of this article was to conduct a scoping review of studies describing the use of restraint and seclusion to manage psychiatric emergencies in clinical settings in Africa. The synthesis of these studies will inform future efforts to improve patient safety and uphold ethical standards in mental health care, including the adaptation and implementation of existing intervention models within African contexts.

## Methods

To conduct this scoping review, we first developed and completed structured searches of five databases: PubMed, Embase, CINAHL, PsycInfo and ProQuest. By including a variety of databases with differing but relevant content areas and research types (e.g., full-length manuscripts, conference abstracts and dissertations/theses), we were able to comprehensively identify literature related to our topic. Search queries were developed in collaboration with a medical librarian and refined to meet the specifications of each database. Search queries included terms for (1) Africa and each of the 47 African countries listed in the WHO Regional Office for Africa listing (WHO, 2025), (2) restraint and seclusion and (3) mental health. Full search queries can be found in Supplementary Appendix A. We performed additional searches using variations of these terms using search engines such as Google and Google Scholar and scanned the references cited in the final included studies to identify gray literature or relevant articles that may have been missed during our structured searches.

Upon completion of our searches', identified works were imported into Covidence, an online platform for organizing systematic reviews. After removal of 104 duplicate records, we identified 376 unique studies that were considered for inclusion in the scoping review (see Figure 1).

The process of screening these records involved two main steps: title/abstract screening and full-text screening. Prior to screening, we developed eligibility criteria to guide inclusion or exclusion of the studies, and these criteria were further refined and finalized

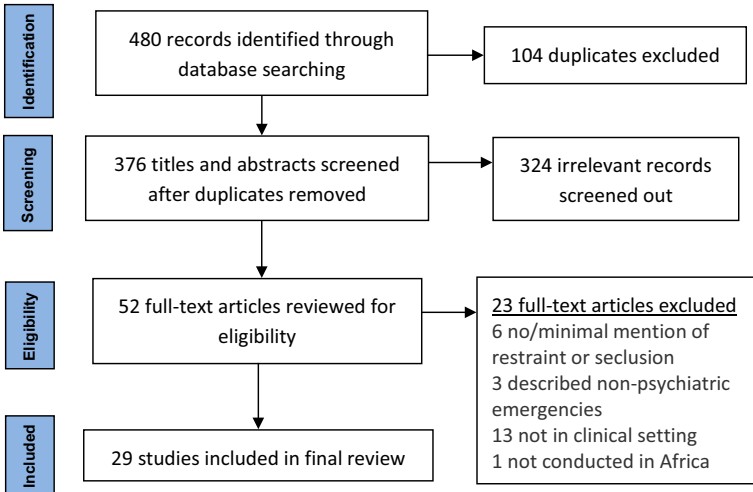

**Figure 1.** Review flowchart.

during the screening process. Inclusion criteria included: studies conducted in Africa, focused on the use of restraint and seclusion for psychiatric emergencies and was conducted in a hospital, clinic or other professional treatment setting. Exclusion criteria included records without data (e.g., commentaries, letters to the editor); prior systematic reviews or literature reviews; case studies or case series with fewer than 10 participants due to concerns about generalizability; studies not presented in English; studies not conducted in Africa (e.g., conducted with African diaspora in other settings); studies with non-African people in African settings (e.g., missionaries, tourists); studies focused on drug and medicine action only (e.g., physiology, drug interactions, biochemistry, genetics); studies in nonclinical or nonprofessional community settings such as traditional healers or religious healers; studies focused on medical confusion, delirium, epilepsy or disorientation with purely medical or neurological (i.e., not psychiatric) etiology; and animal studies.

During title and abstract screening, we reviewed each record to determine whether the study was clearly irrelevant or otherwise ineligible for inclusion. Two authors independently reviewed each record and voted on its advancement for full-text screening. Disagreements between the two raters were resolved in team meetings until a team consensus was reached. Studies advanced to full-text screening were reviewed in detail by two team members who voted on their inclusion in the final review, and once again, disagreements were resolved by the full team until consensus was reached. Studies deemed eligible for the final review during full-text screening were assigned for data extraction by one team member, with review and quality check by a second team member, using structured extraction tables. Extracted data included study nation and setting, topic and objectives, description of participants, data analysis and methods, key findings and author-described implications for future research, practice and policy.

## Results

The sample size of the 29 final included studies varied greatly, ranging from 8 to 572 participants. Of the 29 included studies, seven studies were conducted in South Africa (Mayers et al., 2010; Ramlall et al., 2010; Luckhoff et al., 2013; van Wijk et al., 2014;

Chiba and Subramaney, 2015; Kalula and Petros, 2016; Umubyeyi et al., 2020), six in Tunisia (Jouini et al., 2017; Abdelghaffar et al., 2018; Daoud et al., 2018; Cherif et al., 2019; Messedi et al., 2020a; Maatouk et al., 2022), three in Nigeria (Oyelade and Ayandiran, 2018; Aluh et al., 2022, 2024), three in Ethiopia (Alem et al., 2002; Belete, 2017; Ng et al., 2023b), three in Egypt (Fawzy, 2015; Mahmoud, 2017; El-Sayad, 2018), two in Ghana (Read et al., 2009; Arias et al., 2016), one in Zimbabwe (Sebit et al., 1998), one in Lesotho (Ntsaba and Havenga, 2007), one in Uganda (Coneo et al., 2020), one in Malawi (Barnett et al., 2018) and one in Morocco (Azizi et al., 2023). These studies consisted of 14 quantitative (Sebit et al., 1998; Ramlall et al., 2010; Luckhoff et al., 2013; Chiba and Subramaney, 2015; Kalula and Petros, 2016; Belete, 2017; Jouini et al., 2017; Mahmoud, 2017; Abdelghaffar et al., 2018; Barnett et al., 2018; Cherif et al., 2019; Messedi et al., 2020a, 2020b; Maatouk et al., 2022; Azizi et al., 2023), 10 qualitative (Ntsaba and Havenga, 2007; Read et al., 2009; van Wijk et al., 2014; Arias et al., 2016; Daoud et al., 2018; Oyelade and Ayandiran, 2018; Umubyeyi et al., 2020; Aluh et al., 2022, 2024; Ng et al., 2023b) and 5 mixed method studies (Alem et al., 2002; Mayers et al., 2010; Fawzy, 2015; El-Sayad, 2018; Coneo et al., 2020).

Regarding study populations, participants included patients (Sebit et al., 1998; Ntsaba and Havenga, 2007; Mayers et al., 2010; Luckhoff et al., 2013; van Wijk et al., 2014; Chiba and Subramaney, 2015; Fawzy, 2015; Belete, 2017; Jouini et al., 2017; Mahmoud, 2017; Abdelghaffar et al., 2018; Barnett et al., 2018; Umubyeyi et al., 2020; Aluh et al., 2022; Azizi et al., 2023), medical caregivers such as doctors and nurses (Alem et al., 2002; Ramlall et al., 2010; Daoud et al., 2018; El-Sayad, 2018; Oyelade and Ayandiran, 2018; Cherif et al., 2019; Coneo et al., 2020; Messedi et al., 2020a; Maatouk et al., 2022; Aluh et al., 2024), or mixed groups (Read et al., 2009; Arias et al., 2016; Ng et al., 2023a). See Table 1 for additional details on the included studies.

### Study design and topic

Studies deployed various forms of interventions, including educational interventions for caregivers (El-Sayad, 2018; Cherif et al., 2019), policy-based interventions (Mayers et al., 2010; Barnett et al., 2018; Azizi et al., 2023) and multidisciplinary team-based intervention (Luckhoff et al., 2013; Chiba and Subramaney, 2015; Daoud et al., 2018), focused on reducing restraint or seclusion practices in

**Table 1.** Overview of included studies

| Citation | Country | Sample | Type of analysis | Topic |
|---|---|---|---|---|
| Abdelghaffar et al. (2018) | Tunisia | 52 clinically stable patients hospitalized for their first psychotic episode | Cross-sectional quantitative analysis | Investigation of patient traumatic experiences and PTSD |
| Alem et al. (2002) | Ethiopia | 9 psychiatrists, 9 psychiatry residents, 34 psychiatric nurses | Mixed-methods analysis of attitudinal data | Ethical and legal Implications of restraint, seclusion and involuntary hospitalization in psychiatric emergencies |
| Aluh et al. (2022) | Nigeria | 19 male and 11 female service users in four different focus groups | Inductive, thematic qualitative analysis to derive insights from focus group discussions | Experiences and perceptions of service users in psychiatric facilities regarding restraint and seclusion during emergencies |
| Aluh et al. (2024) | Nigeria | 16 doctors and 14 nurses who had practiced for at least 1 year in the selected psychiatric hospitals | Qualitative thematic analysis of semi-structured interviews | Use of coercive practices, specifically restraint and seclusion, during psychiatric emergencies in Nigeria |
| Arias et al. (2016) | Ghana | 14 Prophets and staff at nine Christian prayer camps in Ghana, and 36 staff within Ghana's three public psychiatric hospitals | Qualitative analysis using the constant comparative method of semistructured interviews with prayer camp and biomedical staff | Collaboration between prayer camp staff and biomedical mental health providers in Ghana to address mental health treatment challenges |
| Azizi et al. (2023) | Morocco | Records of 200 cases in the psychiatric emergency department | Descriptive quantitative analysis of the patient cases | Relationship between the use of involuntary treatment like physical restraint and PTSD symptoms |
| Barnett et al. (2018) | Malawi | Records of 419 psychiatric inpatients | Quantitative multivariate logistic regression analysis | Factors associated with seclusion of patients suffering from mental health disorders and ways to reduce its use to prevent adverse physical and psychological consequences |
| Belete (2017) | Ethiopia | 400 outpatients diagnosed with bipolar disorder | Quantitative, cross-sectional descriptive statistics, multivariate and binary logistic regression | Exploring the factors associated with the use of physical restraint among patients with bipolar disorders |
| Cherif et al. (2019) | Tunisia | Medical staff from psychiatric and non-psychiatric departments (n = 34) | Quantitative, descriptive, transversal study. | Experiences of restraint by medical staff in different hospital departments, implications for care |
| Chiba and Subramaney (2015) | South Africa | Review of records of 112 patients who experienced seclusion | Quantitative, retrospective record review | Determining the number of patients secluded, their characteristics and reasons for seclusion |
| Coneo et al. (2020) | Uganda | 90 nurses and support staff in the psychiatric ward who completed a 4-day training program on the management of aggression. A subset of 35 completed a qualitative interview or focus group | Pre-post mixed-methods convergent design to evaluate a training intervention | Explored how staff attitudes toward causes and management of aggression were affected by a 4-day RESPECT training program |
| Daoud et al. (2018) | Tunisia | 29 nurses from two psychiatric units who had experience of using restraint completed semistructured qualitative interviews | Qualitative, cross-sectional, descriptive epidemiological study | Explored experiences of using restraint, impressions of the practice, impact on the patient and clinical relationship, and risks |
| El-Sayad (2018) | Egypt | 80 nurses working in an inpatient psychiatric unit and 104 patients | Mixed methods, descriptive exploratory study including clinical records, qualitative interviews, and a structured questionnaire | To explore psychiatric patients' and nurses' perspectives on the management of psychiatric emergencies |
| Fawzy (2015) | Egypt | Staff and patients at a large psychiatric hospital, including 18 nurses, 15 psychiatry residents, 3 psychiatrists, 2 consultants, 10 psychologists, 10 social workers, 36 service users, 15 family members | Mixed Methods, cross-sectional. Findings thematically grouped according to five rights from the Convention of Rights of Persons with Disabilities (CPRD) | Provider and service user perceptions of observation of human rights in psychiatric care |
| Jouini et al. (2017) | Tunisia | 240 newly admitted patients for psychotic symptoms in a hospital psychiatry department | Quantitative, retrospective, descriptive and comparative study | Epidemiological profile of patients with schizophrenia spectrum and other psychotic disorder |
| Kalula and Petros (2016) | South Africa | 572 adults and adolescent hospital patients, 46 physicians, 159 nurses | Quantitative, descriptive, cross-sectional study | Prevalence of physical restraint use, patient characteristics associated with physical restraint use, and nurses' and doctors' knowledge and perceptions towards the practice |

*(Continued)*

**Table 1.** (*Continued*)

| Citation | Country | Sample | Type of analysis | Topic |
|---|---|---|---|---|
| Luckhoff et al. (2013) | South Africa | 246 adult patients admitted to the acute psychiatric wards and involved in assault and resulting in seclusion, excluding those in substance use detoxification units | Quantitative retrospective review of clinical records of patients | Assessed trends in patient assaults and the use of seclusion in acute admission wards in a psychiatric facility |
| Maatouk et al. (2022) | Tunisia | 30 nurses and orderlies of the hospital psychiatry department | Quantitative descriptive cross-sectional study, questionnaire | Evaluated the knowledge of nurses and orderlies in the practice and monitoring of physical restraint |
| Mahmoud (2017) | Egypt | 96 nurses working in government mental hospitals and psychiatric wards in general hospitals | Quantitative descriptive study, self-administered questionnaire | Psychiatric nurses' attitudes and practices of physical restraint |
| Mayers et al. (2010) | South Africa | Focus groups with 16 service users who had experienced sedation, seclusion and restraint and questionnaire with 43 service users | Mixed methods with qualitative thematic analysis and cross-sectional descriptive analysis of questionnaire data | Perceptions and experiences of service users exposed to sedation, seclusion and restraint |
| Messedi et al. (2020a and 2020b) | Tunisia | 38 doctors and 30 nurses practicing in a hospital psychiatric unit | Quantitative, cross-sectional, descriptive and comparative study using questionnaires | Explored caregivers' experiences, feelings and knowledge regarding the use of physical restraint |
| Ng et al. (2023a and 2023b) | Ethiopia | 18 patients with severe mental illness, 21 caregivers, 14 primary healthcare providers | Qualitative analysis using an interpretative phenomenological approach | Explored the intersection of severe mental illness and trauma exposure during treatment |
| Ntsaba and Havenga (2007) | Lesotho | 11 adult patients with psychosis or bipolar disorder who completed qualitative interviews | Tesch's method of open qualitative coding | Explored patient experiences with seclusion and views of hospital personnel |
| Oyelade and Ayandiran (2018) | Nigeria | 8 psychiatric–mental health nurses working in a hospital setting | Qualitative analysis of focus group discussion, inductive approach of content analysis by Elo and Kyngäs | Experiences with violence and practices of violence management |
| Ramlall et al. (2010) | South Africa | Survey of 36 hospital managers | Descriptive analysis of quantitative survey responses | Impact of the South African Mental Health Care Act No. 17 on hospital's psychiatric inpatient facilities |
| Read et al. (2009) | Ghana | 67 participants: 25 patients, 31 carers, 3 traditional healers, 4 pastors, 1 mallam (respected elder), 3 imams (religious leader) | Qualitative grounded theory approach: observation, analysis of interview and focus group data | Responses to mental illness in rural and perceptions of the roles of family members, churches, shrines, hospitals and clinics |
| Sebit et al. (1998) | Zimbabwe | 95 restrained/secluded patients in an inpatient psychiatric unit | Descriptive quantitative study | Frequency of restraint and seclusion and the characteristics of the secluded/restrained patients |
| Umubyeyi et al. (2020) | South Africa | 10 patients attending a community psychiatric clinic who conducted in-depth interviews | Qualitative descriptive design using content analysis | Patient experiences of seclusion during their admission in a psychiatric hospital |
| van Wijk et al. (2014) | South Africa | 40 inpatients in two mental health facilities who had been admitted for at least 7 days who completed semistructured interviews | Qualitative, phenomenological analysis using Tesch's descriptive method of open coding | Patient perceptions of the factors that contribute to aggressive and violent behavior after admission to a mental health facility |

clinical settings. Others were non-interventional and descriptive (van Wijk et al., 2014; Fawzy, 2015; Jouini et al., 2017; Aluh et al., 2024).

A majority of the studies primarily investigated restraint (Abdelghaffar et al., 2018; Aluh et al., 2022; Azizi et al., 2023; Barnett et al., 2018; Belete, 2017; Cherif et al., 2019; Coneo et al., 2020; Daoud et al., 2018; El-Sayad, 2018; Jouini et al., 2017; Kalula and Petros, 2016; Mahmoud, 2017; Messedi et al., 2020a, 2020b; Ng et al., 2023a; Oyelade and Ayandiran, 2018), whereas fewer studies focused primarily on seclusion (Ntsaba and Havenga, 2007; Ramlall et al., 2010; Luckhoff et al., 2013; Barnett et al., 2018; Umubyeyi et al., 2020), and six studies investigated both topics (Sebit et al., 1998; Mayers et al., 2010; van Wijk et al., 2014; Fawzy, 2015; Coneo et al., 2020; Aluh et al., 2024).

Definitions of coercive measures varied slightly across studies. One study specified that their definition of coercive measures included involuntary admissions, compulsory treatment, seclusion and restraint (Aluh et al., 2022). Among the studies that provided definitions of restraint, many differentiated between mechanical, chemical and physical restraint (van Wijk et al., 2014; Fawzy, 2015; Oyelade and Ayandiran, 2018; Aluh et al., 2022). Definitions of mechanical and physical restraint tended to overlap between studies, with physical restraint defining any physical methods of restricting a patient's movement (Fawzy, 2015; Mahmoud, 2017), while mechanical restraint referred specifically to the immobilization of a patient with a mechanical device (Aluh et al., 2022). Forms of physical restraint included being held down, tied, shackled and chained in place (Arias et al., 2016; Oyelade and Ayandiran, 2018;

**Table 2.** Key findings of the included studies

| Citation | Key findings | Relevance – Implications for care, research, policy |
|---|---|---|
| Abdelghaffar et al. (2015) | Use of restraint and seclusion contribute to trauma, distress and the development of PTSD in patients | Psychological harm caused by coercive practices, urges mental health services to minimize their use and adopt trauma-informed care approaches |
| Alem et al. (2002) | Nurses were hesitant to discuss the diagnosis or side effects of treatment with the patient. Use of restraints as first line of treatment in the majority of cases where behavior was disruptive. May reflect the lack of availability of staff, training and resources to deescalate these situations | Ethical and practical considerations in acute psychiatric care, including patient rights, informed consent and the limits of restraint and seclusion, evaluates best practices and the potential for policy change in countries where mental health legislation is absent or underdeveloped |
| Aluh et al. (2022) | Negative experience of restraint and seclusion, dehumanizing, as punishment, means to control, lack of care; patients viewed restraint necessary for those more disturbed -> downward stigma | How coercion contributes to the stigma surrounding mental health care. It provides insight into how these practices, while sometimes deemed necessary, may deter service users from seeking psychiatric assistance, further isolating them and reinforcing societal prejudices |
| Aluh et al. (2024) | Staff views restraint as necessary and is aware of side effects; sociocultural context, obsolete mental health legislation, staff shortages and attitudes were factors influencing the use of coercion in mental healthcare | Emphasizes the need for legislative and institutional reforms to reduce coercion and promote less restrictive interventions in mental health care |
| Arias et al. (2016) | Collaboration/clash between biomedical and prayer camp staff; different view on causes of mental illness; chains, fasting used in prayer camps | Exemplifies a cultural divide in how restraint and seclusion is viewed in mental health care as biomedical professionals seek to introduce safer, less restrictive practices through medication, yet prayer camp staff rely on chains and fasting |
| Azizi et al. (2023) | Hospitalization without consent, the use of isolation techniques, restraint or the obligation to take treatment are all factors that can be perceived as traumatic; psychotic episode as well as treatment can be traumatic | Emphasis on the traumatic nature of coercive psychiatric treatments, especially for patients already vulnerable due to PTSD; recommendation of trauma-informed care strategies to mitigate harm |
| Barnett et al. (2018) | Prevalence of seclusion: 30.3% Risk factors seclusion: male gender, aggression towards other patients, being presented in mechanical restraints | Provides data to inform risk assessment and targeted interventions to reduce seclusion in psychiatric inpatient settings in Lilongwe, Malawi |
| Belete (2017) | Prevalence of restraint for bipolar: 18% Risk factors restraint bipolar: Having two or more episodes of bipolar disorder, history of aggression, comorbid illness, use of antipsychotic and current use of Khat | Indicates a high level of restraint in bipolar disorder care in Ethiopia, suggests underlying factors associated with restraint and methods to reduce it |
| Cherif et al. (2019) | 82.4% believed restraint was a prevention tool; 72.8% felt a difference in the relationship with their patient postrestraint, and this change was mostly perceived as negative (70.8%) | Understanding current perspectives toward coercive practices is key in creating policies and practices to decrease stigma and improve the quality of care received by patients |
| Chiba and Subramaney (2015) | Younger male patients with psychosis were most likely to be secluded; reasons: 9.1% for their own safety, 40% for physical or verbal aggression | No data in South Africa on rates of seclusion, making it difficult to ascertain the need for and alternatives to seclusion, despite a substantial proportion of morbidity and mortality associated with seclusion |
| Coneo et al. (2020) | RESPECT intervention: attitudes did not change, attribution of aggression afterwards more external | Training in aggression management skills are not available within the Ugandan curriculum for healthcare providers, despite the fact they face aggression regularly in their day-to-day work. However, a brief training alone does not appear adequate to improve attitudes and practices |
| Daoud et al. (2018) | Reasons for restraint: aggression and agitation (75%) to prevent fall (35%); negative emotional impact on staff (negative type of frustration [25%] or lack of feeling [39%]) | Restraint was most often used in this context to protect the environment and patient but can have adverse consequences if not executed safely |
| El-Sayad (2018) | 94% reported not receiving specific training on skills like communication, restraint, limit setting; No hospital guidelines on limit setting technique; All patients were dissatisfied with how limit setting was implemented | Nurses have limited information on limit setting and require training on empathetic, patient-centered techniques. The experience of limit setting for patients can be improved by providing them with explanations or alternative management techniques |
| Fawzy (2015) | Staff views chemical restraint as necessary; Reasons for restraint: refusing treatment, perception as dangerous, involuntary hospitalization; data on the use of restraint not systematically and comprehensively recorded; service users and families feel forgotten by government, legislation to enforce rights but lack of policy to implement which leads to human rights violations | The research displayed that violations and restrictions of basic human rights were present at El Abbassia. More facilities require assessment and there is a need for advocacy to encourage reforms |
| Jouini et al. (2017) | 21.1% of patients with psychotic disorders experienced physical restraint, compared to 8.1% in other patient groups; mostly male, single and unemployed | May be some differences in how seclusion, restraint is used for patients with psychotic differences compared to those without. Providers can and should analyze the clinical and |

*(Continued)*

**Table 2.** (*Continued*)

| Citation | Key findings | Relevance – Implications for care, research, policy |
| --- | --- | --- |
| | | sociodemographic characteristics of newly admitted patients to guide and tailor care for different patient groups |
| Kalula and Petros (2016) | No incident of restraint was found in psychiatry | Restraint use is highly prevalent in general hospital settings and poorly coordinated with limited knowledge by providers on how and when to use restraints, based on hospital policy. Necessary to provide clear guidelines on restraint use |
| Luckhoff et al. (2013) | More male patients are involved in assaults; More male patients are secluded and restrained; the number of assaults and seclusions increased over time | Importance of monitoring assaults and seclusion to improve quality of care in mental health facilities. Gender plays a major role in assault and the practice of seclusion, important to understand the drivers of this, but also how to implement gender differentiated approaches |
| Maatouk et al. (2022) | 50% of staff received training focused on physical restraint; 56.6% ignored the psychological effects of the physical restraint on patients; 73.3% of caregivers informed patients before restraint | There is a need for training in mental health and on the use of physical restraint as well as a need for a physical restraint protocol |
| Mahmoud (2017) | Nurses see restraint as necessary but have insufficient staff and resources for restraint and safety monitoring. Lack of knowledge regarding patient rights. Insufficient documentation. Very low awareness of the emotional impacts of restraint on patients | There is a need for more staff to reduce restraint and to better monitor patient safety, need for more training in alternative nursing practices and in patient rights, need for better documentation |
| Mayers et al. (2010) | Inadequate communication, isolation; excessive/inappropriate use of force, lack of respect for basic human dignity; Seclusion as punishment; sedation was seen as least distressing and used most often | Calls for rights-based care, better communication and respect for patient dignity to minimize distress and prevent human rights violations |
| Messedi et al. (2020a and 2020b) – two conference abstracts Nigeria | Provider descriptions of patients' feelings: anger and dehumanization. The use of restraint was abusive according to 33.8%, the main reason was related to a lack of resources for 30.8%. Restraint leads to patient security. Rationale for restraint: aggressiveness against other persons (78.6%), self-aggressiveness (55.4%) and agitation (62.5%) | The study demonstrates a need to educate staff on the use of physical restraint and to continuously question the practice to avoid abusive restraint and painful experiences |
| Ng et al. (2023a and 2023b) | Negative feedback loop: trauma contributes to worsening of severe mental illness(SMI) in Ethiopia, SMI symptoms and caregiver strain increase risk of trauma exposure | The study highlights the need to recognize trauma as a public health issue in low-resource settings, promote integrated care for trauma and mental illness and implement interventions to reduce stigma and support caregivers |
| Ntsaba and Havenga (2007) | Seclusion as punishment, prison-like, human rights violation, degrading, intense negative feelings; lack of information, support and care | Examines how patients view and interact with seclusion in a hospital setting |
| Oyelade and Ayandiran (2018) | Violence in psychiatric wards toward nurses feel unsafe, powerless, react with intimidation, wish for actively armed military officials, lack of alternatives; negative feelings ("heartbrokenness") | Demonstrates the importance of intervention studies on violence management in African mental healthcare settings to reduce violence in psychiatric settings |
| Ramlall et al. (2010) | Positive effect of policy (Mental Health Care Act No. 17), but lack of sufficient beds, staff and seclusion rooms. Lack of training | This study examines how a mental health act can lead to change but also what further steps, resources and training are needed for policy to effectively initiate change |
| Read et al. (2009) | Chaining, beating and fasting are commonplace in community settings; Psychiatric care is barely available; no negative reports of restraint in psychiatry | Emphasizes the need for locally informed interventions that resonate with community practices and beliefs and looks at all ways mental illness is addressed in rural Ghana to understand the families, patients and provider perspectives |
| Sebit et al. (1998) | Prevalence of seclusion: 8.5% of assaults are common; female patients were more likely to physically assault staff or other patients; reason for seclusion: verbal then physical assault; violence linked to alcohol use | First study that examined the characteristics and use of restraint and seclusion in Zimbabwe and sub-Sahara Africa; understanding how, when and in response to which behavior restraint and seclusion is used is a prerequisite for interventions. Need for training nurses in the prevention and control of violent patients |
| Umubyeyi et al. (2020) | Seclusion as punishment and an abusive, negative emotional experience; Inadequate communication | Understanding the adverse effect seclusion has on patients underscores the need to review practices, policies and procedures regarding the use of seclusion. Open communication between the care providers and the patients should be emphasized during the time of seclusion |
| van Wijk et al. (2014) | Environmental factors significantly contribute to aggression (ward atmosphere, living conditions, staff attitudes) | Both environmental factors and staff attitudes contribute to patient aggression and should be addressed to prevent violence; aggression management strategies should be incorporated into staff training programs at mental health facilities |

Ng et al., 2023a). Chemical restraint was defined as using psycho-tropic or sedative drugs to reduce patient movement or aggressive behavior (Aluh et al., 2022). The definition of seclusion was generally consistent across studies and involved confining a patient to an environment that is controlled and contained (Ntsaba and Havenga, 2007; Chiba and Subramaney, 2015; Umubyeyi et al., 2020). Ntsaba and Havenga specified that seclusion should be accompanied by careful observation by a staff member, and some studies described that seclusion should be in a specific facility or room (Ramlall et al., 2010; van Wijk et al., 2014).

The primary point of divergence regarding coercive measures was their purpose. While most definitions agreed that the main purpose of coercive measures was to control unsafe behaviors (Barnett et al., 2018; Daoud et al., 2018; El-Sayad, 2018; Umubyeyi et al., 2020), some authors argued that coercive measures could also serve therapeutic purposes (Sebit et al., 1998; Maatouk et al., 2022). Barnett et al. (2018), however, emphasized that these measures should only be used for emergency contexts and have no therapeutic value .

The topics investigated in these studies included service users' perspectives on coercive practices in inpatient settings (Ntsaba and Havenga, 2007; Mayers et al., 2010; van Wijk et al., 2014; Umubyeyi et al., 2020; Aluh et al., 2024), as well as caregivers' experiences with physical restraint (Messedi et al., 2020b). Additionally, four studies examined various healthcare workers' perspectives and attitudes toward coercive methods and aggression management (Mahmoud, 2017; Cherif et al., 2019; Coneo et al., 2020; Maatouk et al., 2022).

Some studies explored the frequency of restraint and seclusion, demographic profiles of affected populations and trends in restraint and seclusion (Sebit et al., 1998; Luckhoff et al., 2013; Kalula and Petros, 2016; Belete, 2017; Jouini et al., 2017; Barnett et al., 2018). A few studies identified that restraint and seclusion contributed to patient trauma and posttraumatic stress disorder (PTSD) (Abdelghaffar et al., 2018; Azizi et al., 2023; Ng et al., 2023a). Additionally, two studies examined the ethical, legal and human rights implications related to the use of restraint and seclusion (Alem et al., 2002; Fawzy, 2015).

### *Study quantitative outcomes*

The prevalence of restraint and seclusion varied widely across studies. The highest rate was observed in a study in South Africa, where 83.7% of patients reported experiencing seclusion, although only 14.0% had been restrained (Mayers et al., 2010). In a study of 400 people with bipolar disorders in Ethiopia, 65% had experienced physical restraint at some point (Belete, 2017). Studies in inpatient psychiatric units in Tunisia (Jouini et al., 2017), South Africa (Kalula and Petros, 2016) and Malawi (Barnett et al., 2018) reported rates of restraint and seclusion of 21.1%, 23.0% and 30.3% of patients, respectively. Another study looking at both restraint and seclusion in Zimbabwe found a much lower prevalence of 8.5% (Sebit et al., 1998).

Restraint and seclusion were frequently used to manage agitation, physical and verbal aggression and to prevent self-harm (Sebit et al., 1998; Chiba and Subramaney, 2015; Messedi et al., 2020a, 2020b). Physical restraint was also used as a preventive measure, or to enable involuntary treatment (Fawzy, 2015; El-Sayad, 2018; Cherif et al., 2019). Male gender, young age, psychosis, comorbid illnesses, aggressive behavior and being presented in restraints as well as contextual factors such as being unemployed or single were identified as significant risk factors for the use of restraint or seclusion (Sebit et al., 1998; Luckhoff et al., 2013; Chiba and Subramaney, 2015; Belete, 2017; Jouini et al., 2017; Barnett et al., 2018).

Khat use was identified as a predictor of restraint in one study (Belete, 2017), but substance use was not associated with increased use of restraint in a second study (Barnett et al., 2018). Two studies identified restraint and seclusion as predictors of patient trauma and PTSD (Abdelghaffar et al., 2018; Azizi et al., 2023) and patients viewed.

Among quantitative studies examining provider and nurse perceptions, restraint and seclusion were viewed by providers as a necessary treatment and the first and most preferred response to patient agitation (Alem et al., 2002; El-Sayad, 2018). One study limited provider training on restraint and seclusion and limited understanding of policies dictating their use (Kalula and Petros, 2016). Two studies showed poor documentation and monitoring of patient safety (Kalula and Petros, 2016; Mahmoud, 2017) and two studies showed low provider understanding of the emotional impacts of restraint and seclusion on patients (El-Sayad, 2018; Maatouk et al., 2022). Only one study described an intervention aimed at changing attitudes and practices related to restraint and seclusion, where the team observed short-term change in provider attitudes but no change in their intention to use restraint and seclusion in future practice (Coneo et al., 2020). See Table 2 for additional descriptions of the key findings of each study.

### *Synthesis of qualitative, quantitative and mixed-methods outcomes*

From the patients' perspective, restraint and, especially, seclusion were often reported to be used as punishment or as a means of control (Ntsaba and Havenga, 2007; Mayers et al., 2010; Umubyeyi et al., 2020; Aluh et al., 2022). Healthcare professionals reported how the lack of resources, staff and knowledge of alternative practices contributed to aggression and the use of restraint (Alem et al., 2002; Ramlall et al., 2010; van Wijk et al., 2014; Messedi et al., 2020a, 2020b; Aluh et al., 2024). Patients experienced restraint and seclusion as dehumanizing, leading to strong negative emotional reactions and the development of PTSD (Ntsaba and Havenga, 2007; Mayers et al., 2010; Abdelghaffar et al., 2018; Umubyeyi et al., 2020; Aluh et al., 2022; Azizi et al., 2023; Ng et al., 2023a). Seclusion was described as prison-like (Ntsaba and Havenga, 2007). The lack of adequate communication, support and care from staff was often described by patients (Ntsaba and Havenga, 2007; Mayers et al., 2010; Umubyeyi et al., 2020; Messedi et al., 2020a; Aluh et al., 2022).

Healthcare professionals frequently viewed restraint and seclusion as necessary in the management of aggression and violence, which are often common in psychiatric facilities and pose an occupational hazard (Sebit et al., 1998; van Wijk et al., 2014; Oyelade and Ayandiran, 2018). Several studies reported that restraint and seclusion practices place an emotional burden on staff, leading to feelings of guilt, "heartbrokenness" or frustration (Mahmoud, 2017; Oyelade and Ayandiran, 2018). Two studies looking at training in restraint found that 50% and 94% of staff in these settings did not receive formal professional training in restraint and seclusion (El-Sayad, 2018; Maatouk et al., 2022). One study described an intervention that targeted staff attitudes without significantly changing them (Coneo et al., 2020).

Several studies have problematized human rights and legal issues regarding restraint and seclusion. In one study, restraint was perceived as abusive by healthcare workers in 33.8% of cases (Messedi et al., 2020b). Other studies describe its insufficient documentation (Fawzy, 2015; Mahmoud, 2017) and the lack of policies and guidelines to guide and limit restraint and seclusion (Fawzy, 2015; El-Sayad, 2018) (Figure 2).

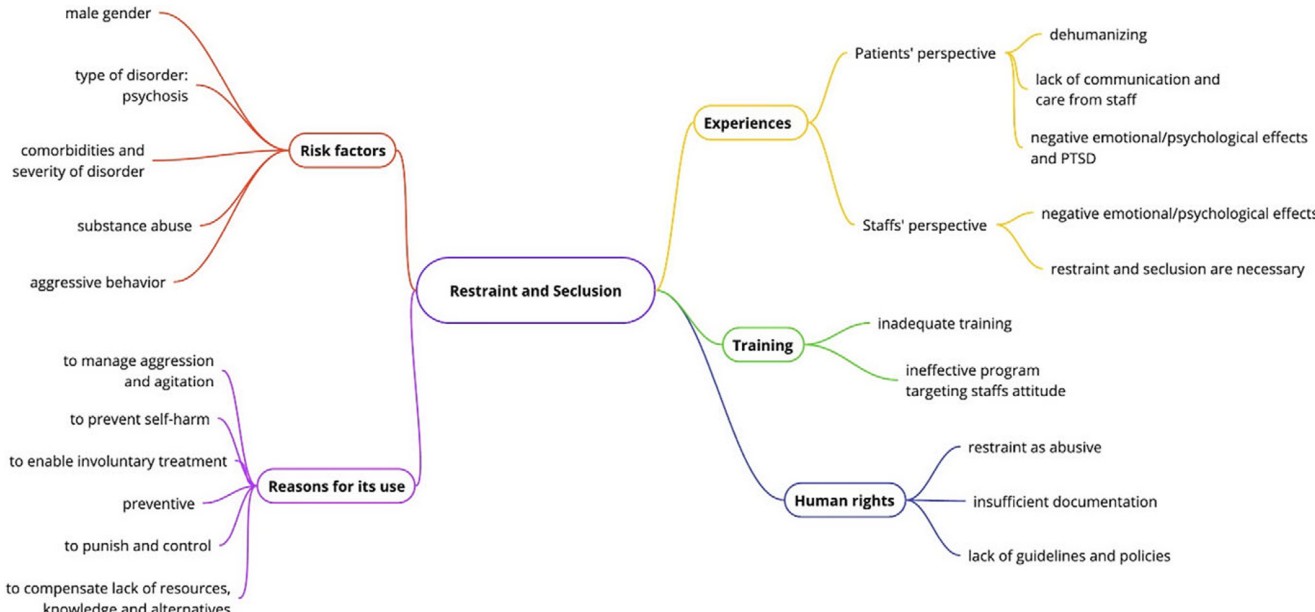

**Figure 2.** Summary of key themes reflected in the included studies.

## Implications described in the included studies

The studies included in this scoping review highlighted implications for care, policy and research regarding the use of restraint and seclusion in African hospital settings. Many studies emphasized the human rights abuses and emotional trauma associated with these practices, which often deter patients from seeking psychiatric care (Abdelghaffar et al., 2018; Aluh et al., 2022; Azizi et al., 2023; Cherif et al., 2019; Fawzy, 2015; Messedi et al., 2020b; Ng et al., 2023a; Ntsaba and Havenga, 2007). A few studies identified that ethical and practical considerations related to restraint and seclusion by emphasizing patient rights, and ensuring informed consent through open communication and education (Alem et al., 2002; Mayers et al., 2010; Daoud et al., 2018; Umubyeyi et al., 2020).

Most studies proposed reforms, with some studies suggesting increased documentation and monitoring of restraint and seclusion (Luckhoff et al., 2013; Chiba and Subramaney, 2015; Jouini et al., 2017; Mahmoud, 2017; Barnett et al., 2018). A few studies advocated for behavioral interventions to address violence and aggression management to reduce reliance on these practices (Jouini et al., 2017; Barnett et al., 2018; Oyelade and Ayandiran, 2018). Many studies highlighted the importance of training healthcare providers in patient-centered care, limit setting and proper management of psychiatric emergencies (Sebit et al., 1998; Ntsaba and Havenga, 2007; Mayers et al., 2010; Kalula and Petros, 2016; El-Sayad, 2018; Cherif et al., 2019; Coneo et al., 2020; Messedi et al., 2020a; Maatouk et al., 2022).

Many studies pointed to policy implications, especially the need for institutional and legislative reforms in regions with absent or underdeveloped mental health laws but must be complemented by systemic and behavioral changes (Alem et al., 2002; Ramlall et al., 2010; Fawzy, 2015; Belete, 2017; Cherif et al., 2019; Umubyeyi et al., 2020; Aluh et al., 2024). The sole intervention study in this review evaluated a 4-day training program seeking to reduce restraint and seclusion but had little impact (Coneo et al., 2020). Another study examined a much more far-reaching policy change, the South African Mental Health Care Act of 2002, which stipulated that regional and district hospitals must admit and treat mental health care users for 72 h before transferring them to a psychiatric

hospital (Ramlall et al., 2010). Although this policy was described as an effort to improve human rights among people struggling with mental illness, its implementation put considerable stress on the healthcare system because of inadequate infrastructure and staff and may have actually led to increases in the inhumane use of restraint and seclusion outside of psychiatric settings (Ramlall et al., 2010). The studies emphasized the need for systemwide reforms that extend beyond policy change to include provider training, clear guidelines on the safe use of restraint and seclusion, and improved regulation and oversight, as well as dedication of new resources to support the safe and effective implementation of these strategies (Mayers et al., 2010; Kalula and Petros, 2016; Mahmoud, 2017; El-Sayad, 2018; Messedi et al., 2020b; Maatouk et al., 2022).

## Discussion

This scoping review presents findings from 29 heterogenous studies conducted in Africa exploring the use of restraint and seclusion to manage psychiatric emergencies in clinical settings. The studies explored provider, patient and caregiver perspectives on these practices. Nominally, restraint and seclusion were used to ensure patient and environmental safety; however, multiple challenges were identified in the extent and way these practices were used. Both patients and providers acknowledged that restraint and seclusion were overused across multiple settings and were harmful to the physical and emotional well-being of both patients and providers. Patients identified examples of inappropriate and inhumane uses of restraint and seclusion, such as for punitive purposes.

Regarding efforts to reduce the use of restraint and seclusion and to minimize harm in its use, some key takeaways from this scoping review are that there is limited knowledge by providers on aggression management and crisis de-escalation. Furthermore, many settings lacked clear standards on when these practices should be used and how to use them in a way that maximizes effectiveness and safety. This included lack of continual monitoring of patient safety, not clearly communicating to patients why these practices were being used and when they would be discontinued, and poor ongoing assessment of symptoms to monitor safety and

discontinue restraint and seclusion as quickly as possible. The lack of training and clear standards led to obvious ethical and human rights violations in many settings.

Many of the studies suggested reforms through policy; however, several also advocated that policy changes must be coupled with improved documentation, training and other measures to ensure more ethical practices within psychiatric emergencies. The studies also highlight significant gaps in infrastructure, training and policy, contributing to the widespread use of restraint and seclusion. In contrast to higher-income countries, African settings often lack the necessary systemic support, with insufficient training, resource shortages and a lack of standardized guidelines. Furthermore, the absence of culturally informed interventions and comprehensive documentation limits the ability to implement effective reforms. Future research will not only spread awareness of the issues at a national level but also encourage greater targeted reform improving the quality of patient care in psychiatric hospitals.

Future research should focus on developing, adapting and implementing culturally tailored, least-restrictive interventions for managing psychiatric emergencies. Moreover, longitudinal studies and intervention trials are essential to assess the effectiveness of interventions and reforms, ensure ethical practices and improve mental health outcomes. Expanding this body of work will be vital in addressing the complex intersection of mental health care, human rights and resource limitations in Africa.

This scoping review has several limitations. First, we only included studies published in English, which may have led to the omission of important studies conducted in other languages. Additionally, although we searched multiple academic databases and additional searches to identify gray literature, we may have missed unpublished studies and published work that was not included in these databases. Furthermore, case studies and case series with fewer than 10 participants were excluded, as were studies focusing on nonhospital settings, such as faith-based or traditional healing contexts. Future studies may seek to explore these types of research. Many of the included studies relied on participant self-report and, given the sensitivity of the topic, biases in reporting likely influenced the results, potentially leading to underreporting of harmful practices. Finally, the broad scope of this review, covering a vast continent and complex topic, limited our ability to provide detailed descriptions of individual study outcomes in specific African settings. Although we conducted a thorough search inclusive of all 47 countries on the continent as defined by the WHO, the final list of 29 studies represented only 11 nations in Africa, and findings should not be generalized beyond those settings. This dearth of research on restraint and seclusion practices throughout much of the continent also points to the clear value of future studies on the topic.

## Conclusions

The topic of psychiatric emergencies and the associated use of restraint and seclusion in Africa is of critical importance, given the unique challenges facing many health systems on the continent, including resource constraints, stigma surrounding mental health and gaps in the availability of trained mental health professionals. Restraint and seclusion practices, while sometimes necessary to ensure safety, raise ethical and human rights concerns that demand urgent attention. This scoping review has illuminated challenges associated with restraint and seclusion across various African contexts, including its overuse, unsafe use and use for punitive

purposes. We underscore the need for culturally sensitive and patient-centered care while highlighting the importance of reforming current practices to prioritize safety, dignity and the minimization of trauma. The findings serve as a foundation for guiding policy changes, advocacy efforts, intervention research and improvements in training for mental health professionals.

**Open peer review.** To view the open peer review materials for this article, please visit http://doi.org/10.1017/gmh.2025.10052.

**Supplementary material.** The supplementary material for this article can be found at http://doi.org/10.1017/gmh.2025.10052.

**Data availability statement.** Data or details of the underlying processes of this scoping review are available upon request from the corresponding author.

**Author contribution.** All authors contributed to the design, analysis, interpretation of findings and manuscript writing, and approved the final submitted version of the work.

**Financial support.** This research was supported by a Career Development Award from the NIH National Institute of Mental Health (K08 MH124459). We also acknowledge support received from the grant, "Sociobehavioral Sciences Research to Improve Care for HIV Infection in Tanzania" (D43 TW009595).

**Competing interests.** The authors have no competing interests to declare.

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
