## [Reviewer Report]

This is a well-structured and timely scoping review that addresses a critical gap in the literature on psychiatric emergency management in African clinical settings. The topic is of high relevance, however, the manuscript can benefit from clearer methodological justification, deeper synthesis of findings, and expanded discussion on global and regional policy implications.

Comments to the authors

1. Include terminology consistency and definitions early

In Lines: 68–75, the definition of restraint and seclusion appears mid-way through the background; they should be defined upfront.

2. Expand the global context

In Lines: 86–96, “In high-income global settings, efforts have included mandated training… It will be crucial to implement similar strategies in other global settings, including throughout the continent of Africa.” This paragraph is too brief and under-referenced. The authors can expand by comparing Africa’s situation with countries implementing specific reforms.

3. Strengthen the rationale for a Scoping Review

In the Lines: 97–101 “The objective of this manuscript is to conduct a scoping review of studies describing the use of restraint and seclusion…” The rationale for using a scoping review method is not well-articulated. The authors can add a brief explanation to strengthen the rationale.

4. Data extraction approach needs more depth

In Lines: 159–161, “Studies deemed eligible… were assigned for data extraction by one team member, with review and quality check by a second team member…” This is descriptive but lacks analytic insight. The authors can add details on how the extracted data were organized.

5. Clarify the Scope of ‘Africa’

Under the results (Lines: 163–181) “Of the 29 included studies, seven studies were conducted in South Africa... one in Morocco.” The findings may be biased due to more concentration of studies in only a few African countries. The authors can add a clarifying statement in the Introduction or Methods acknowledging that regional representation is uneven and discuss how this may limit the generalizability of the review findings across Africa.

6. Balance between quantitative and qualitative data

Lines: 174–181, Although the studies consisted of 14 quantitative and 11 qualitative, the quantitative findings are under-discussed compared to qualitative narratives. The authors can expand the Results section with comparative metrics.

7. Policy recommendations should be actionable

In Lines: 319–321, “Many studies pointed to policy implications… must be complemented by systemic and behavioral changes…” Policy recommendations are too broad, specify using some examples. On note line 323, “Expand on this.” Did the authors intend to expand on this?

8. Limitations section needs expansion

Under discussion, “Furthermore, the absence of culturally informed interventions and comprehensive documentation limits the ability to implement effective reforms.” This is phrased as a discussion point rather than a structured limitations section. The authors can add it to the limitations paragraph.

9. The limitations of the scoping review (e.g., language restriction to English, exclusion of grey literature in some regions, limited generalizability due to geographic skew) should be explicitly listed and discussed.

---

## [Reviewer Report]

• This is a well-written and timely paper addressing a highly relevant topic in African mental health care.

• The authors are encouraged to clarify their database selection. Specifically, why was African Journals Online (AJOL) not included in the search strategy, given its relevance to regional research?

• There is some inconsistency in the referencing format. At times, only surnames are used, while in other instances, both surnames and initials appear. Please ensure uniformity throughout the manuscript.

• The statement: “We also examine strategies that have been implemented to reduce restraint and seclusion, which may inform future efforts to improve patient safety and uphold ethical standards in mental health care” may be misleading. The review did not include studies explicitly focused on implementation of strategies, and this should be accurately reflected in the discussion.

• While the paper rightly identifies the need for context-appropriate solutions, it is important to acknowledge that effective strategies already exist. Rather than reinventing the wheel, the focus should be on adapting and implementing existing models within African contexts.

---

## [Editor Report]

Thank you for the submission of your manuscript to Cambridge Prisms: Global Mental Health. As both reviewers indicate, your manuscript makes and important contribution to the field of Global Mental Health. Both reviewers do, however make recommendations for strengthening of the manuscript. Kindly attend to these comments and make a re-submission.

---

## [Reviewer Report]

Thank you for your comprehensive responses and review.

The revised manuscript shows clear improvements in context, methodology, and policy relevance. All my previously noted issues, including the rationale for the scoping review, terminology consistency, and limitations, have been adequately addressed. However, a few minor stylistic and grammatical issues remain that do not hinder understanding. They are more of editorials, which can be dealt with in final proofreading.